# CalMol: Disentangled Causal Graph LLM for Molecular Relational Learning

## Abstract

Molecular Relational Learning (MRL), focused on understanding interactions between molecular pairs, is essential for drug design with both structural and textual information, *i.e.*, molecular structures and textual documents. However, most existing MRL methods assume identical molecular distributions, failing in the ubiquitous real-world scenarios involving new drugs with distribution shift, which is mainly due to the reason that they heavily reply on *variant* correlations between structures and texts regarding interactions that tend to change when new drugs or molecules come. To solve this problem, we investigate zero-shot MRL, by leveraging *invariant* relationships between molecular texts and structures w.r.t interactions in the course of time, which is largely unexplored in the literature and is highly non-trivial with the following challenges: 1) How to disentangle molecular structure components between each pair that intrinsically determine interactions, and address potential structural distribution shift issues for new drugs? 2) How to align molecular structures with semantic textual information to achieve invariant molecular relation predictions for new drugs? To tackle these challenges, we propose a novel Causally Disentangled Invariant Graph Large Language Model for Molecular Relational Learning (**CalMol**), capable of exploiting invariant molecular relationships to predict interactions for new drugs. In particular, we propose *Causal Molecule Substructure Disentanglement* to capture the invariant well-recognized substructure pair for a specific molecule interaction. Then, we propose *Molecule Structure and Property aware LLM Alignment* to use molecule (with invariant substructure)-textual property pair to align structure information to semantic information, and utilize them together to guide the interaction prediction. On this basis, LLM can also provide further explanations. Extensive experiments on **qualitative** and **quantitative** tasks including **7 datasets** demonstrate that our proposed **CalMol** achieves advanced performance on predicting molecule interactions involving new molecules.

## 1 Introduction

Molecular Relational Learning (MRL), aiming to understand interactions between molecular pairs, plays a pivotal role in advancing biochemical research with both structural and textual information, *i.e.*, molecular structures and documents. For example, in drug discovery, it is crucial to consider the interactions between molecules, based on both their structural and textual properties (Chang & Ye, 2024; Jin et al., 2020; Dou et al., 2022).

As the development of new molecules, such as drugs, accelerates, the challenge of evaluating interactions involving these novel compounds becomes increasingly critical (Zhu et al., 2024). However, most existing MRL methods assume identical molecular distributions and struggle in situations where limited information is available about new molecules, including their relationships with previously known compounds. A key problem is that they tend to rely on either variant molecular structures, which might dominate the molecular space (Yang et al., 2022), or associated textual information (Dou et al., 2022), leading to difficulties in prediction, especially when confronted with distribution shift and evolving information.

To solve this problem, we investigate **zero-shot MRL**, *i.e.*, predicting relations involving new molecules, by leveraging *invariant* relationships between molecular texts and structures w.r.t interac-

tions in the course of time. This is a largely unexplored area in the literature and presents several highly non-trivial challenges:

- How to disentangle structural components between molecule pair that inherently determine interactions and mitigate potential distribution shifts in molecular structures for new drugs?

- How to align molecular structures with semantic textual information to achieve invariant molecular relation predictions for new drugs?

To address these challenges, we propose a novel **C**ausally Disentangled Invariant GrAph Large **L**anguage Model for **Mol**ecular Relational Learning (**CALMOL**), capable of exploiting invariant molecular relationships for predicting interactions involving new drugs. Our method leverages the complementary strengths of Graph Neural Networks (GNNs) for molecule structural learning and Large Language Models (LLMs) for text processing, information retrieval and integration (Lyu et al., 2023; Li et al., 2024), aiming to provide a more comprehensive understanding of molecular interactions across diverse scenarios, particularly when dealing with both known and novel molecules.

Particularly, we propose *Causal Molecule Substructure Disentanglement* to identify invariant, well-recognized motif pairs that govern molecule interactions. This is achieved by decomposing molecules into chemically coherent motifs and applying causal constraint along with Gumbel-Sigmoid Reparameterization masking method to disentangle causal motif-interaction information from the entangled molecular embeddings. Given the causal motif pairs obtained from the above module, we introduce *Molecule Structure and Property aware LLM Alignment* to align molecular structural information (with invariant substructures) to semantic information, using the structure-property pairs, and further incorporate them to guide interaction predictions. The motivation behind this is that different motifs within a molecule may be responsible for various molecular properties. By focusing on the causal motif pairs, we can encourage the LLM to identify relevant properties and make predictions based on the causal motif and property, as depicted in figure 1. This approach also enables the LLM to offer additional explanations for the interactions. Empirical validation across both **qualitative** and **quantitative** tasks including **7 datasets** demonstrate that our proposed **CALMOL** achieves advanced performance on predicting molecule interactions involving new molecules. Detailed ablation studies further verify our designs. The contributions of this paper are summarized as follows:

- We study Graph LLM for zero-shot MRL, which is largely unexplored, by proposing novel **C**ausally Disentangled Invariant GrAph Large **L**anguage Model for **Mol**ecular Relational Learning (**CALMOL**), capable of exploiting invariant molecular relationships for predicting interactions involving new drugs.

- We propose two modules: i) Causal Molecule Substructure Disentanglement to capture the invariant well-recognized substructure pair for a specific molecule interaction; and ii) Molecule Structure and Property aware LLM Alignment to use molecule (with the obtained invariant substructure)-textual property pair to align structure information to semantic information, and guide interaction prediction. On this basis, LLM can further provide meaningful explanations.

- Extensive experiments on qualitative and quantitative tasks including 7 datasets demonstrate that our proposed **CALMOL** achieves advanced performance on predicting molecule interactions involving new molecules. [1]

## 2 PRELIMINARY

### 2.1 PROBLEM FORMALIZATION

Molecular Relational Learning (MRL) seeks to predict the interaction (either classification or regression) between a pair of molecules used together. Since new molecules are continuously being developed and emerging, learning interactions involving these novel molecules poses a significant challenge. To address this, we focus on this largely unexplored area by framing it as a zero-shot learning problem. During the training phase, interactions are observed among a set of known molecules. In the inference phase, the goal is to predict interactions involving either a new molecule paired with a known one or between two entirely new molecules. Formally, we define the task as follows:

---

[1]We provide codes of our paper in the anonymous link.

**Definition 1** *(Zero-shot Molecular Relational Learning)*

*Let $\mathcal{M}$ denote the set of all molecules, $\mathcal{M}_{new} \subset \mathcal{M}$ the set of novel molecules, and $\mathcal{I}$ the set of interaction outcomes, where $\mathcal{I}$ can represent either qualitative classification labels or quantitative regression values. Formally, the zero-shot molecular relational learning task is to learn a mapping $F : (\mathcal{M}_{new} \times \mathcal{M}) \cup (\mathcal{M} \times \mathcal{M}_{new}) \rightarrow \mathcal{I}$, where $F$ maps a molecule pair $(u, v) \in ((\mathcal{M}_{new} \times \mathcal{M}) \cup (\mathcal{M} \times \mathcal{M}_{new}))$ to an interaction outcome $i \in \mathcal{I}$, which can be either a qualitative interaction type (for a classification task) or a quantitative value (for a regression task).*

## 2.2 CAUSAL MOLECULE SUBSTRUCTURE AS A BRIDGE

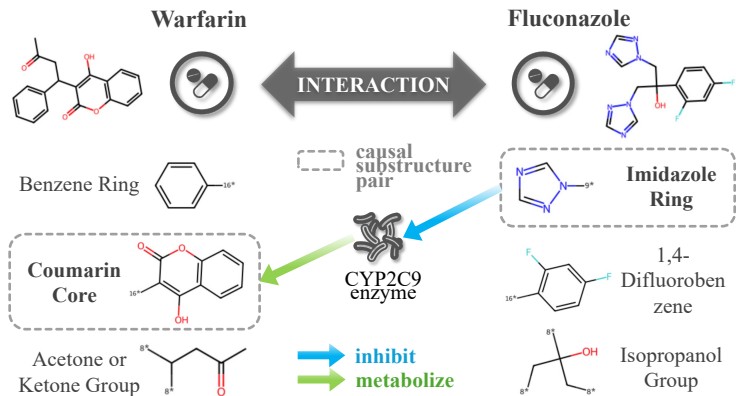

Figure 1: **MRL is driven by causal substructure pair and related property.** The interaction between these two drugs is primarily driven by the imidazole ring in fluconazole, which inhibits the CYP2C9 enzyme responsible for metabolizing the coumarin core in warfarin. This inhibition slows down the breakdown of warfarin, causing its concentration to increase in the bloodstream, which heightens the risk of excessive anticoagulation and bleeding.

To harness the structural modeling capabilities of Graph Neural Networks (GNNs) alongside the information integration and textual processing strengths of Large Language Models (LLMs)—which complement each other in Molecular Relational Learning (MRL)—we propose using core, well-recognized molecular substructures as a bridge to integrate these two powerful models for interaction prediction. Specifically, we outline the process of abstracting these substructures in Section 3.1, and explain how they serve as a bridge between the GNN and LLM, facilitating interaction prediction in Sections 3.2 and 3.3.

## 3 METHODOLOGY

In this section, we introduce our **CALMOL** in detail. Since it is difficult to directly decompose a molecule and extract the causal part for predicting interaction between a pair of molecules, we utilize a GNN-based model to learn to extract the causal part when encoding a molecule structure at first, which then serves as the graph encoder and causal substructure pair extractor in our Graph LLM **CALMOL**. Specifically, we introduce the proposed causal molecule substructure disentangling module in section 3.1, then the molecule substructure & property aware LLM alignment module in 3.2, and finally sum up the entire training and inference procedure of **CALMOL** in section 3.3.

## 3.1 CAUSAL MOLECULE MOTIF-INTERACTION DISENTANGLING

**Molecule Decomposition.** Previous work in Graph Learning often abstracts core subgraphs by selecting nodes and edges of high importance based on specific regularizers (Li et al., 2022; Wu et al., 2022). However, subgraphs obtained in this manner may appear fragmented, especially in molecular graphs, where the resulting subgraphs often lack the chemical coherence of meaningful substructures. To address this, we propose using the BRICS algorithm (Degen et al., 2008) to decompose molecules

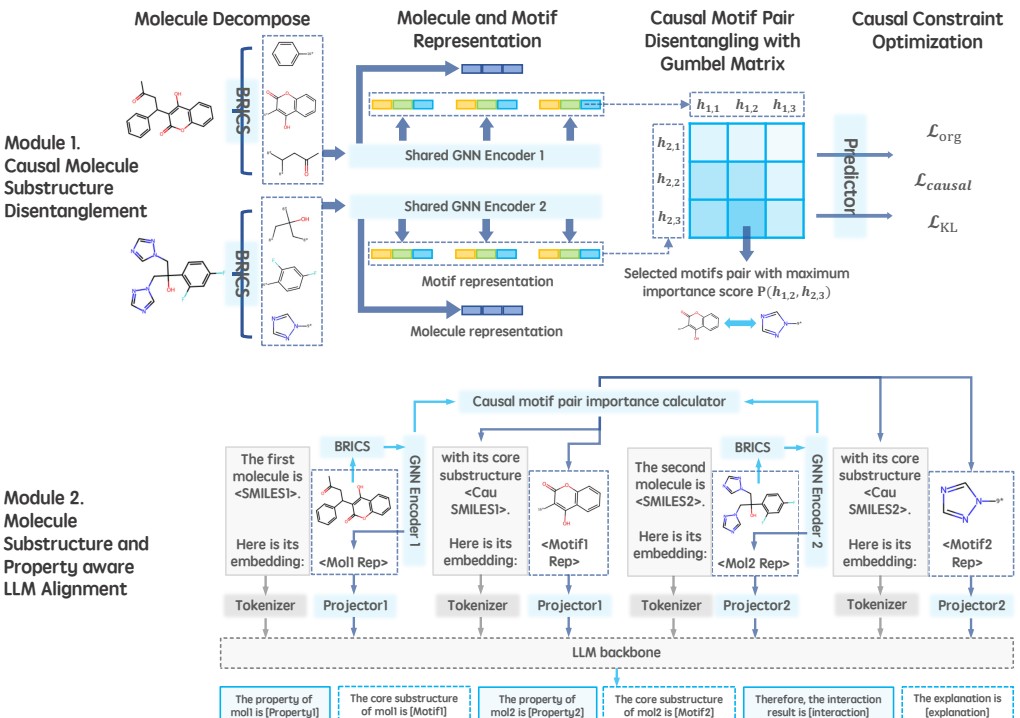

Figure 2: **CALMOL** framework. As for a molecule pair, the Causal Molecule Substructure Disentanglement module first identify invariant, well-recognized motif-interaction pair that inherently governs interaction, by decomposing molecules into chemically coherent motifs and applying causal constraint along with Gumbel-Sigmoid Reparameterization masking method to disentangle causal motif-interaction information. Based on the trained GNN encoders and causal motif pair calculator from above, the Molecule Structure and Property aware LLM Alignment module aligns molecular structural information (with causal motifs) to semantic information, using the structure-property pairs, and further incorporate them to guide interaction predictions. The detailed training procedure is in figure 3.

into chemically meaningful motifs. Since BRICS cleaves bonds based on a predefined set of chemical reactions, the resulting motifs retain chemical integrity and are more easily recognized by LLMs. Given molecular pairs $M_1$ and $M_2$, we fragment their respective molecular graphs $\mathcal{G}_1$ and $\mathcal{G}_2$ into motif sets $\{\mathcal{U}_i\}, i \in [1, N_1]$ and $\{\mathcal{V}_j\}, j \in [1, N_2]$, where $N_1$ and $N_2$ denote the total number of motifs corresponding to $\mathcal{G}_1$ and $\mathcal{G}_2$, respectively.

**Molecule and Motif Representation.** Given each original molecule $\mathcal{G}_1$ and $\mathcal{G}_2$, and their respective motifs $\mathcal{U}_i$ and $\mathcal{V}_j$, we first derive atom-level representations using shared GNN encoder. Specifically, the embedding are obtained as follows:

$$\mathbf{E}_1 = \text{GNN}_1(\mathcal{G}_1), \mathbf{E}_2 = \text{GNN}_2(\mathcal{G}_2), \mathbf{E}_{\mathcal{U}_i} = \text{GNN}_1(\mathcal{U}_i), \mathbf{E}_{\mathcal{V}_j} = \text{GNN}_2(\mathcal{V}_j). \quad (1)$$

Note that for the drug-drug interaction task, we use the same GNN encoder for both molecules and their corresponding motifs. However, in the solute-solvent interaction task, we employ two different GNN encoders for the solute and solvent molecules, respectively, since their structures can vary significantly. Next, we compute the graph representations $\mathbf{h}_1$ and $\mathbf{h}_2$ for the molecules $\mathcal{G}_1$ and $\mathcal{G}_2$, as well as $\mathbf{h}_{\mathcal{U}_i}$ and $\mathbf{h}_{\mathcal{V}_j}$ for each motif, using the Set2Set readout function (Vinyals et al., 2015).

**Causal Motif Pair Disentangling with Gumbel Matrix.** Considering the influence of micro interaction between all possible motif pairs $\mathbb{P} = \{(\mathcal{U}_i, \mathcal{V}_j)\}, (i \in [1, N_1], j \in [1, N_2])$ on macro molecule relation, we propose to separate the causal motif pair $C = (\mathcal{C}_1, \mathcal{C}_2)$ and the shortcut motif pair $S = (\mathcal{S}_1, \mathcal{S}_2)$ from $\mathbb{P}$ in latent space. In detail, we first form a motif-interaction representation matrix $\mathbf{T} \in \mathbb{R}^{N_1 \times N_2 \times 2d}$, in which $\mathbf{T}_{ij} = \mathbf{h}_{\mathcal{U}_i} || \mathbf{h}_{\mathcal{V}_j}$ is the concatenation of two motifs' embedding

for pair $(\mathcal{U}_i, \mathcal{V}_j)$. Based on it, we disentangle causal part $\mathbf{C}$ and spurious part $\mathbf{S}$ from $\mathbf{T}$ by masking it with a differentiable Gumbel matrix $\mathbf{\Lambda} \in \mathbb{R}^{N_1 \times N_2}$ as follow:

$$\mathbf{C}_{ij} = \lambda_{ij} \mathbf{T}_{ij} + (1 - \lambda_{ij})\epsilon, \quad \mathbf{S}_{ij} = (1 - \lambda_{ij})\mathbf{T}_{ij}, \tag{2}$$

in which $\lambda_{ij} \sim \text{Bernoulli}(p_{ij})$, $\epsilon \sim \mathcal{N}(\mu, \sigma^2)$ is the noisy motif-interaction feature, and $\mu, \sigma^2$ denote the mean and variance of $\mathbf{T}$. To be specific, sampling $\lambda_{ij}$ from Bernoulli distribution, which is a non-differentiable operation, can be avoid through *Gumbel-Sigmoid Reparameterization* (Jang et al., 2016; Maddison et al., 2016) as follows:

$$\lambda_{ij} = \text{sigmoid}\left(\frac{\log(p_{ij}/(1-p_{ij})) + g}{\tau}\right), \quad g = -\log(-\log(u)), \tag{3}$$

where $p_{ij}$ is the Bernoulli probability, $\tau$ is the temperature parameter, $g$ is the Gumbel noise and $u \sim \text{Uniform}(0, 1)$. Mention that, since $p_{ij}$ indicates the probability of $\lambda_{ij}$ being 1, we regard it as the importance/probability of the motif pair $(\mathcal{U}_i, \mathcal{V}_j)$ being a causal part in final prediction, and learn it from the motif-interaction representation with MLP:

$$p_{ij} = \text{MLP}(\mathbf{T}_{ij}), \quad \mathbf{P} \in \mathbb{R}^{N_1 \times N_2}. \tag{4}$$

**Causal Constraint Optimization.** We compel the above section to disentangle the causal pair by optimizing the following objective function:

$$\mathcal{L} = \mathcal{L}_{org}(Y, \hat{Y}) + \mathcal{L}_{causal}(Y, \hat{Y}_{\mathbf{C}}) + k \cdot \mathcal{L}_{KL}(Y_r, \hat{Y}_{\mathbf{S}}), \tag{5}$$

$\mathbf{C}$ as the causal part in $\mathbf{T}$, is guaranteed by $\mathcal{L}_{causal}(Y, \hat{Y}_{\mathbf{C}})$ to solely determine the final prediction. On the other hand, $\mathbf{S}$ is expected to contain no predictive information by optimizing $\mathcal{L}_{KL}$, which forces the distribution of predictions from $\mathbf{S}$ to resemble a random distribution.

### 3.2 MOLECULE SUBSTRUCTURE AND PROPERTY AWARE LLM ALIGNMENT

This section is motivated by the observation that different molecular substructures are linked to distinct properties, which, in turn, influence molecular interactions. For instance, in a drug molecule, the primary structure responsible for therapeutic effects typically governs its key properties and interactions with other molecules. Meanwhile, secondary functional groups, such as those that impart hydrophilicity, may affect solubility but are not directly involved in the molecular reaction. Understanding these specific roles allows the model to more effectively align substructures with molecular properties, enhancing predictions in molecular pair interactions.

**Molecule Structure Embedding.** Given the original molecule graph pair, we first utilize the frozen GNN encoders to obtain atom-level representations of the original molecule graphs $\mathbf{E}_1$ and $\mathbf{E}_2$, as well as the atom-level representations of their motifs, denoted as $\mathbf{E}_{\mathcal{U}_i}$ for $i \in [1, N_1]$ and $\mathbf{E}_{\mathcal{V}_j}$ for $j \in [1, N_2]$. Next, we compute the importance matrix $\mathbf{P} \in \mathbb{R}^{N_1 \times N_2}$ for motif pairs $(\mathcal{U}_i, \mathcal{V}_j)$ based on motif-interaction representations using the corresponding frozen importance calculator MLP, as described in Equations 2, 3, and 4. Given the significant difference in importance scores within $\mathbf{P}$, we select the motif pair $(\mathcal{U}_c, \mathcal{V}_c)$ corresponding to the highest score $p_{max}$, and use their embeddings $\mathbf{E}_{\mathcal{U}_c}$ and $\mathbf{E}_{\mathcal{V}_c}$ for further processing in the LLM pipeline.

**Molecule Representation Projector.** Given the atom-level representations $\mathbf{E}_1$, $\mathbf{E}_2$, $\mathbf{E}_{\mathcal{U}_c}$ and $\mathbf{E}_{\mathcal{V}_c}$, the next step is to map them into the backbone LLM's hidden space using the projectors $f_{\text{pro1}}$ and $f_{\text{pro2}}$. These projectors take essential responsibility for aligning GNN language $\mathbf{E}_1$, $\mathbf{E}_2$, $\mathbf{E}_{\mathcal{U}_c}$ and $\mathbf{E}_{\mathcal{V}_c}$ into corresponding encodings $\mathbf{Q}_1$, $\mathbf{Q}_2$, $\mathbf{Q}_{\mathcal{U}_c}$ and $\mathbf{Q}_{\mathcal{V}_c}$ that are compatible with the LLM. Following the approach of state-of-the-art vision-language models, we implement $f_{\text{pro1}}$ and $f_{\text{pro2}}$ using Querying Transformers (Q-Formers), as in the works of Li et al. (2023a) and Dai et al. (2023). Specifically, the encodings are defined as

$$\mathbf{Q}_1 = [q_i^1] = f_{\text{pro1}}(\mathbf{E}_1), \mathbf{Q}_{\mathcal{U}_c} = [q_i^u] = f_{\text{pro1}}(\mathbf{E}_{\mathcal{U}_c}), \tag{6}$$

$$\mathbf{Q}_2 = [q_i^2] = f_{\text{pro2}}(\mathbf{E}_2), \mathbf{Q}_{\mathcal{V}_c} = [q_i^v] = f_{\text{pro2}}(\mathbf{E}_{\mathcal{V}_c}), \quad i \in [1, l]$$

with $l$ representing the number of learnable query tokens in the Q-Former. The projectors, built on the BERT architecture, incorporate a cross-attention module between the self-attention and feed-forward

layers, which enables complex alignment between molecule structural information, especially the causal substructure pair, and molecule property. It also accommodates flexible input graph embedding sizes, with learnable query token dimensions adjustable to match the token embedding size of the language model. This architecture enhances effective interactions between multi-modal molecular information during LLM inference.

**SMILES Tokenization.** SMILES tokenization helps distinguish molecules in a pair by providing a unique, linear representation for each molecule. This structured notation not only maintains molecule identity but also embeds their sequential order clearly. We utilizes SMILES for its widespread use and precision, allowing the molecule's information to link effectively with the LLM's biochemical knowledge. Additionally, the BRICS decomposition method in Section 3.1 enables us to derive SMILES for molecular substructures, further aiding the model in recognizing the key components of each molecule. Finally, **CALMOL** directly input the four SMILES strings of $\mathcal{G}_1, \mathcal{G}_2, \mathcal{U}_c, \mathcal{V}_c$ into the backbone LLM, leveraging the encoder to capture their tokenized representations $\mathbf{R}_1, \mathbf{R}_2, \mathbf{R}_{\mathcal{U}_c}, \mathbf{R}_{\mathcal{V}_c}$, which ensures accurate molecule identification.

**Choice of LLM.** Following MolTC (Fang et al., 2024), **CALMOL** utilizes Galactica, a decoder-only transformer based on the OPT architecture, as its core language model. Trained on an extensive dataset of scientific texts, Galactica excels in biochemistry, particularly in interpreting molecular sequences like SMILES and molecular property from various documents. This specialized information allows it to effectively capture key properties related to molecular structures and interactions. By leveraging Galactica's huge biochemical knowledge repositories, strong biochemical information integration and inferential capabilities, **CALMOL** can analyze and interpret the contextual interactions between two basic molecular token sets, $\{\mathbf{R}_1, \mathbf{Q}_1, \mathbf{R}_{\mathcal{U}_c}, \mathbf{Q}_{\mathcal{U}_c}\}$ and $\{\mathbf{R}_2, \mathbf{Q}_2, \mathbf{R}_{\mathcal{V}_c}, \mathbf{Q}_{\mathcal{V}_c}\}$.

---

**Prompt for Molecule Structure and Property Aware Alignment**

**Input Prompt:** The first molecule is `<SMILES1>`, `<GraEmb1>`, with its core substructure `<CauSMILES1>`, `<CauGraEmb1>`, and the second molecule is `<SMILES2>`, `<GraEmb2>`, with its core substructure `<CauSMILES2>`, `<CauGraEmb1>`. Please provide the biochemical properties of the two molecules one by one.

**Target Answer:** The properties of the first molecule are `[Property1]`, and the properties of the second molecule are `[Property2]`.

---

### 3.3 TRAINING AND INFERENCE PROCEDURE OF **CALMOL**

To conclude the above modules, we introduce the complete training and inference procedure of **CALMOL** in this section.

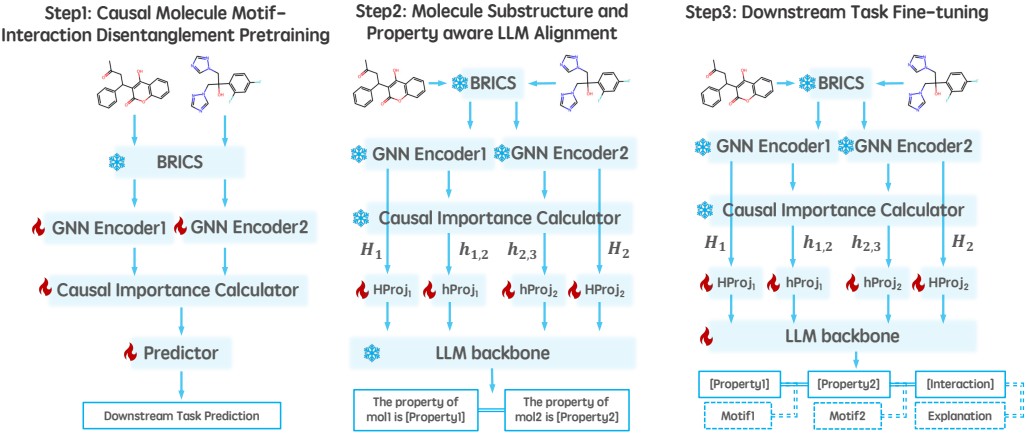

Figure 3: Three steps in training procedure.

**Training Procedure.**    The training process is divided into three steps:

1. The first step involves pretraining the Causal Molecule Motif-Interaction Disentangling module, as outlined in Section 3.1. Directly training the full hybrid Graph-LLM model to embed molecules and extract causal motifs is highly resource-intensive and may impede the model's ability to concentrate on specific structural causal learning tasks. To address this, we first pretrain this module independently, ensuring it effectively captures structural information and causal motif interactions. This enhances the overall performance of **CALMOL** when integrated into the subsequent training phases. After the pretraining of the module in Section 3.1, we freeze the GNN encoders from Equation 1 and the importance calculator from Equation 4, to retain their ability on structural modeling and causal motif pair identifying, then incorporate them into our Graph LLM, **CALMOL**.

2. The second step involves training the Molecule Substructure and Property Aware LLM Alignment module, as detailed in Section 3.2. In this step, we adopt the frozen GNN encoder and causal importance calculator obtained from the first step, along with the frozen backbone LLM. The focus here is solely on training the projectors. This setup allows us to refine the alignment of (causal) structures and properties within the molecular embeddings without overloading the model with the complexities of the full architecture. By isolating the projectors, we ensure that they effectively map the GNN outputs to the LLM space, improving downstream performance.

3. The third step involves fine-tuning the integrated model on specific downstream tasks, such as DDI classification or SSI regression prediction. In this step, beyond further refining the projectors to better suit the downstream task, we primarily focus on fine-tuning the LLM to adapt to task-specific output formats. For instance, the LLM is trained to generate specific classification sentences for DDI prediction or to produce precise numerical values for SSI regression. This targeted fine-tuning ensures that the LLM effectively interprets and outputs results that align with the requirements of each downstream task, enhancing both the accuracy and interpretability of predictions.

Prompt for molecule structure and property aware alignment is shown in Section 3.2, and prompts for downstream tasks fine-tuning are shown as below.

---

**Prompt for Drug-Drug Interaction**

**Input Prompt:**    The first molecule is `<SMILES1>`, `<GraEmb1>`, with causal substructure `<CauSMILES1>`, `<CauGraEmb1>`, and the second molecule is `<SMILES2>`, `<GraEmb2>`, with causal substructure `<CauSMILES2>`, `<CauGraEmb1>`. What are the side effects of these two drugs?

**Target Answer:**    The first molecule has causal substructure `[CauSMILES1]` and property `[Property1]`. The second molecule has causal substructure `[CauSMILES2]` and property `[Property2]`. Therefore, the drug1 may increase the photosensitizing activities of drug2. Explanation based on their causal substructures and properties is that `[Explanation]`.

---

**Prompt for Solute-Solvent Interaction**

**Input Prompt:**    The first molecule is `<SMILES1>`, `<GraEmb1>`, with causal substructure `<CauSMILES1>`, `<CauGraEmb1>`, and the second molecule is `<SMILES2>`, `<GraEmb2>`, with causal substructure `<CauSMILES2>`, `<CauGraEmb1>`. What is the solvation Gibbs free energy of these two molecules?

**Target Answer:**    The first molecule has causal substructure `[CauSMILES1]` and property `[Property1]`. The second molecule has causal substructure `[CauSMILES2]` and property `[Property2]`. Therefore, the solvation Gibbs free energy of these two molecules is `[VALUE]`

---

**Inference Procedure.**    The inference process go through our designed and tuned models following the path: $\{\text{SMILES1}, \mathcal{G}_1, \text{SMILES2}, \mathcal{G}_2\} \rightarrow$ **[BRICS]** $\rightarrow$ **[GNN encoder]** $\rightarrow$ **[importance calculator]** $\rightarrow \{\text{SMILES}_1, \mathbf{E}_1, \text{SMILES}_{\mathcal{U}_c}, \mathbf{E}_{\mathcal{U}_c} \text{ and } \text{SMILES}_2, \mathbf{E}_2, \text{SMILES}_{\mathcal{V}_c}, \mathbf{E}_{\mathcal{V}_c}\} \rightarrow$ **[LLM backbone]** $\rightarrow \{\mathbf{R}_1, \mathbf{Q}_1, \mathbf{R}_{\mathcal{U}_c}, \mathbf{Q}_{\mathcal{U}_c} \text{ and } \mathbf{R}_2, \mathbf{Q}_2, \mathbf{R}_{\mathcal{V}_c}, \mathbf{Q}_{\mathcal{V}_c}\}$.

## 4 EXPERIMENT

In this section, we conduct extensive experiments on real-world molecular relation learning datasets, including drug-drug interaction (DDI) and solute-solvent interaction (DDI) prediction tasks, to verify the design of our method in comparisons with state-of-the-art Graph-based, ML-based, and LLM-based MRL baselines.

### 4.1 EXPERIMENTAL SETTING

**Construction of zero-shot datasets** Inspired by Zhu et al. (2024), we construct each dataset for zero-shot molecular relational learning (MRL) by dividing the set of molecules, $\mathcal{M}$, into three disjoint sets: $\mathcal{M}_{\text{train}}$, $\mathcal{M}_{\text{val}}$, and $\mathcal{M}_{\text{test}}$. Denote the total number of interactions as $\mathcal{S} = \{(u, i, v) : u, v \in \mathcal{M}, i \in \mathcal{I}\}$. Based on this molecule split, the training, validation, and test sets are defined as follows:

- $\mathcal{S}_{\text{train}} = \{(u, i, v) \in \mathcal{S} : u, v \in \mathcal{M}_{\text{train}}\}$;
- $\mathcal{S}_{\text{val}} = \{(u, i, v) \in \mathcal{S} : (u \in \mathcal{M}_{\text{train}} \cup \mathcal{M}_{\text{val}}) \wedge (v \in \mathcal{M}_{\text{train}} \cup \mathcal{M}_{\text{val}}) \wedge (u, i, v) \notin \mathcal{S}_{\text{train}}\}$;
- $\mathcal{S}_{\text{test}} = \{(u, i, v) \in \mathcal{S} : (u \in \mathcal{M}_{\text{train}} \cup \mathcal{M}_{\text{test}}) \wedge (v \in \mathcal{M}_{\text{train}} \cup \mathcal{M}_{\text{test}}) \wedge (u, i, v) \notin \mathcal{S}_{\text{train}}\}$.

In this way, we ensure that novel molecules remain unseen during **CALMOL**'s training process. The statistics of each zero-shot datasets are summarized in Table 1.

Table 1: Dataset statistics.

| Task | Dataset | Original Dataset | | | Molecule Split | | | Zero-shot Dataset | | |
|------|---------|------|------|------|------|------|------|------|------|------|
| | | $\mathcal{M}_1$ | $\mathcal{M}_2$ | Pairs | $\mathcal{M}_{\text{train}}$ | $\mathcal{M}_{\text{val}}$ | $\mathcal{M}_{\text{test}}$ | Pairs $_{\text{train}}$ | Pairs $_{\text{val}}$ | Pairs $_{\text{test}}$ |
| DDI Classification | ZhangDDI | 542 | 543 | 95245 | 435 | 65 | 44 | 60780 | 19812 | 14653 |
| | ChChMiner | 871 | 905 | 32735 | 767 | 115 | 77 | 20759 | 6948 | 5099 |
| | DeepDDI | 1704 | 1704 | 313220 | 1363 | 204 | 137 | 195893 | 68852 | 48475 |
| SSI Regresssion | FreeSolv | 560 | 1 | 560 | 448 | 45 | 68 | 447 | 45 | 68 |
| | CompSol | 442 | 259 | 3548 | 50 | 33 | 50 | 2681 | 342 | 525 |
| | Abraham | 1038 | 122 | 6091 | 988 | 44 | 66 | 5016 | 410 | 665 |
| | CombiSolv | 1415 | 309 | 8780 | 1521 | 67 | 102 | 7111 | 744 | 925 |

**Baselines** We perform a thorough evaluation using a variety of baseline methods as benchmarks, including coventional GNN based models, non-GNN ML based models, and state-of-the-art LLMs. In the DDI task, baselines include CIGIN (Pathak et al. (2020)), MHCADDI (Deac et al. (2019)), DeepDDI (Ryu et al. (2018)), SSI-DDI (Nyamabo et al. (2021)), CGIB (Lee et al. (2023a)), CMRL (Lee et al. (2023b)), and DSN-DDI (Li et al. (2023b)), while SSI tasks utilize D-MPNN (Vermeire & Green (2021)), CIGIN, CGIB, and CMRL. Across all downstream tasks, LLM-based methods like MolTC (Fang et al. (2024)) is implemented for further comparison.

**Metrics** For qualitative tasks, we employ prediction Accuracy and AUC-ROC (Area Under the Receiver Operating Characteristic curve) as comparative metrics, while for quantitative tasks, MAE (Mean Absolute Error) and RMSE (Root Mean Square Error) are utilized as the standards.

**Training Details** During the LLM training process, the choice of our optimizer is AdamW (Loshchilov (2017)), configured with a weight decay of 0.05. Our learning rate schedule starts with linear warm-up to accelerate initial training, then shifts to a cosine decay that gently reduces the learning rate, allowing for smoother fine-tuning of the model. Moreover, we implement LoRA via the Open Delta library (Ding et al. (2022)) and PEFT library (Mangrulkar et al. (2022)). It is configured with a rank of 16 and is implemented on Galactica's layers with a sequence of q-proj, v-proj, out-proj, fc1 and fc2, as described in (Liu et al. (2023)).

For pretraining causal GNN module, we apply graph encoder instantiated by the three-layer GINE (Hu et al. (2019)). In parallel, the projectors are initialized with Sci-BERT, an encoder-only transformer pretrained on scientific texts (Beltagy et al. (2019)). The cross-attention layers are randomly initialized. For the LLM-based baselines, the backbone LLMs are fine-tuned on task-relevant datasets to ensure

a fair comparison. Accuracy is achieved when predictions include only the correct interaction details, with no mention of alternative interactions.

As for the training epochs, we typically perform 10 epochs for alignment module and 100 epochs for fine-tuning each datasets and test on the best epoch. Most datasets can reach best performance within 20 epochs. The optimizer and learning rate scheduler, as outlined in the preceding paragraph, are configured consistently for alignment-training and fine-tuning.

## 4.2 QUALITATIVE ZERO-SHOT DDI RESULTS

As for qualitative zero-shot DDI classification task, Table 2 demonstrates **CALMOL**'s outstanding performance in terms of both accuracy and AUC-ROC on qualitative zero-shot DDI tasks, in comparison with a majority of baseline methods.

Table 2: Comparative performance of various methods in qualitative zero-shot DDI tasks. The best-performing methods are in **bold**, while the second-best methods are underlined for emphasis.

| Setting | Model | ZhangDDI | | ChChMiner | | DeepDDI | |
|---|---|---|---|---|---|---|---|
| | | Accuracy↑ | AUC-ROC↑ | Accuracy↑ | AUC-ROC↑ | Accuracy↑ | AUC-ROC↑ |
| GNN Based | CIGIN | $67.26_{\pm1.39}$ | $\underline{72.12_{\pm1.05}}$ | $79.65_{\pm0.29}$ | $80.78_{\pm0.02}$ | $73.68_{\pm0.10}$ | $84.49_{\pm0.87}$ |
| | SSI-DDI | $54.73_{\pm0.03}$ | $55.82_{\pm0.06}$ | $59.56_{\pm0.37}$ | $63.64_{\pm0.39}$ | $58.01_{\pm0.04}$ | $62.14_{\pm0.11}$ |
| | DSN-DDI | $55.04_{\pm0.59}$ | $60.48_{\pm0.06}$ | $62.35_{\pm0.28}$ | $67.48_{\pm0.41}$ | $67.12_{\pm0.50}$ | $74.52_{\pm0.21}$ |
| | CMRL | $67.03_{\pm1.15}$ | $70.56_{\pm1.65}$ | $78.72_{\pm1.14}$ | $\underline{82.66_{\pm2.31}}$ | $75.39_{\pm1.58}$ | $\mathbf{84.67_{\pm1.00}}$ |
| | CGIB | $\underline{69.26_{\pm0.42}}$ | $\mathbf{74.68_{\pm0.74}}$ | $\underline{79.88_{\pm0.35}}$ | $81.75_{\pm0.91}$ | $\underline{76.38_{\pm0.12}}$ | $84.15_{\pm0.02}$ |
| ML Based | DeepDDI | $57.36_{\pm0.73}$ | $51.44_{\pm1.05}$ | $63.28_{\pm1.70}$ | $54.42_{\pm1.80}$ | $60.55_{\pm0.27}$ | $56.09_{\pm0.98}$ |
| | MHCADDI | $63.48_{\pm0.82}$ | $64.03_{\pm1.72}$ | $72.80_{\pm0.89}$ | $68.30_{\pm1.80}$ | $69.68_{\pm0.47}$ | $72.40_{\pm0.57}$ |
| LLM Based | MolTC | $66.44_{\pm0.38}$ | $64.77_{\pm0.39}$ | $79.26_{\pm1.87}$ | $71.34_{\pm3.84}$ | $69.69_{\pm0.00}$ | $74.06_{\pm0.12}$ |
| | **CALMOL** | $\mathbf{70.69_{\pm0.24}}$ | $67.32_{\pm0.88}$ | $\mathbf{81.12_{\pm0.56}}$ | $\mathbf{82.81_{\pm1.05}}$ | $\mathbf{77.85_{\pm0.63}}$ | $78.92_{\pm0.36}$ |

An in-depth analysis of the experimental results is provided as follows: The proposed **CALMOL** surpasses all benchmark methods in accuracy, showing a consistent improvement of over 1% across various categories of baselines. Notably, it achieves accuracy rates exceeding 70% on each dataset, a performance level that none of the benchmark models were able to reach. Furthermore, the method we propose shows substantial and broad enhancements compared to LLM-based alternatives. This comprehensive improvement affirms the unique strengths of our model architecture in this domain.

Another point worth noting is the weaker and more fluctuating AUC-ROC metrics of LLM-based models for DDI tasks. This phenomenon is explained by their evaluation based on discrete target text matching, effectively interpreting results as a binary 0/1 classification. In contrast, GNN and ML-based approaches leverage probabilistic predictions as computation, providing them with a performance edge in AUC-ROC comparisons. This existing limitation does not diminish the overall promise of our method in the LLM-based framework, highlighting its substantial areas of strength.

## 4.3 QUANTITATIVE ZERO-SHOT SSI RESULTS

Table 3 highlights the dominant regression performance of our model in quantitative SSI tasks. **CALMOL** consistently outperforms other SSI baseline models on all zero-shot datasets, as evaluated by MAE and RMSE metrics. These experimental results highlight the robustness of our model in handling quantitative tasks, particularly in zero-shot scenarios. This underscores the model's efficacy and strong generalization capabilities to perform well on previously unseen data structures.

Additionally, it is worth emphasizing that our proposed approach demonstrates an extraordinary enhancement over the innovative LLM-based model MolTC, with an over 50% reduction in MAE and RMSE on average. When compared to GNN-based models, this figure stands at an approximately 10% to 20%. These statistics showcase the considerable strength of **CALMOL**'s method in the optimal deployment of LLMs' generalized proficiency and flexibility.

Table 3: Comparative performance of various methods in quantitative zero-shot SSI tasks. The best-performing methods are in **bold**, while the second-best methods are underlined for emphasis.

| Setting | Model | FreeSolv | | Abraham | | CompSol | | CombiSolv | |
|---------|-------|----------|---|---------|---|---------|---|-----------|---|
| | | MAE ↓ | RMSE ↓ | MAE ↓ | RMSE ↓ | MAE ↓ | RMSE ↓ | MAE ↓ | RMSE ↓ |
| GNN Based | CIGIN | $0.557_{\pm0.334}$ | $0.856_{\pm0.092}$ | $0.467_{\pm0.016}$ | $0.779_{\pm0.008}$ | $0.472_{\pm0.038}$ | $0.856_{\pm0.059}$ | $0.502_{\pm0.029}$ | $0.829_{\pm0.029}$ |
| | D-MPNN | $0.703_{\pm0.279}$ | $0.884_{\pm0.328}$ | $0.528_{\pm0.012}$ | $0.775_{\pm0.031}$ | $0.660_{\pm0.085}$ | $1.001_{\pm0.131}$ | $0.559_{\pm0.042}$ | $0.846_{\pm0.042}$ |
| | CMRL | $0.510_{\pm0.041}$ | $0.862_{\pm0.035}$ | $0.390_{\pm0.022}$ | $0.660_{\pm0.021}$ | $0.435_{\pm0.038}$ | $0.669_{\pm0.057}$ | $0.428_{\pm0.024}$ | $0.727_{\pm0.029}$ |
| | CGIB | $1.825_{\pm0.305}$ | $2.257_{\pm0.327}$ | $1.835_{\pm0.317}$ | $2.602_{\pm0.529}$ | $1.219_{\pm0.158}$ | $1.507_{\pm0.176}$ | $1.464_{\pm0.288}$ | $1.983_{\pm0.354}$ |
| LLM Based | MolTC | $2.776_{\pm0.365}$ | $3.836_{\pm0.661}$ | $0.690_{\pm0.061}$ | $1.156_{\pm0.111}$ | $0.576_{\pm0.065}$ | $1.083_{\pm0.140}$ | $0.701_{\pm0.038}$ | $1.130_{\pm0.106}$ |
| | **CALMOL** | $\mathbf{0.478}_{\pm0.102}$ | $\mathbf{0.782}_{\pm0.091}$ | $\mathbf{0.335}_{\pm0.028}$ | $\mathbf{0.608}_{\pm0.018}$ | $\mathbf{0.341}_{\pm0.048}$ | $\mathbf{0.612}_{\pm0.096}$ | $\mathbf{0.349}_{\pm0.005}$ | $\mathbf{0.622}_{\pm0.016}$ |

## 5 RELATED WORK

**Traditional computational methods** for Molecular Relational Learning (MRL), particularly those based on Graph Neural Networks (GNNs) and Machine Learning (ML), primarily focus on molecule structure modeling. These approaches, however, are often limited by their reliance on spurious structural correlations and the inability to incorporate textual property information, which could provide critical insights. For example, GNN-based methods such as DDIPrompt (Wang et al. (2024)), which utilizes graph prompt learning, and SSI-DDI (Nyamabo et al. (2021)), focusing on substructure-substructure interactions for drug-drug interaction (DDI) prediction, model only molecular structures. Other works like SA-DDI (Yang et al. (2022)) and DSIL-DDI(Tang et al. (2023)) propose domain-invariant substructure interaction learning, addressing explainability and generalizability, yet still fall short in integrating complementary textual information or handling distribution shifts.

On the other hand, recent **language model based approaches** for MRL leverage the powerful contextual understanding of large language models. However, these methods are prone to spurious correlations in the textual data, which can lead to hallucination and inaccurate predictions. For example, Zhu et al. (2024) explore zero-shot DDI prediction guided by textual drug description, while MolTC (Fang et al. (2024)) investigates comprehensive molecular relational modeling using language models and structure embedding from GNN. Despite their innovation, these approaches still lack robust handling of structural information inherent in molecular graphs.

Our work addresses these limitations by studying Graph LLMs for zero-shot MRL, an area largely unexplored in the literature. We propose a hybrid approach that leverages the complementary strengths of GNNs for precise structure modeling and causal part disentangling, as well as LLMs for integrating rich external information, enabling a more comprehensive and robust understanding of molecular interactions.

## 6 CONCLUSION

Most existing Molecular Relational Learning (MRL) methods assume identical molecular distributions, which fall short in the ubiquitous scenarios involving new drugs with different distributions. In this paper, we study zero-shot MRL to predict molecular relations for new molecules, by proposing a novel Causally Disentangled Invariant Graph Large Language Model (**CALMOL**), designed to leverage invariant molecular relationships for predicting interactions with new drugs. Specifically, we first propose Causal Molecule Substructure Disentanglement, designed to identify and capture the invariant, well-recognized substructure pairs critical for specific molecular interactions. Building on this, we propose Molecule Structure and Property Aware LLM Alignment, to align molecular structures, specifically those with invariant substructures, with their corresponding textual properties to integrate structural and semantic information effectively. This alignment enhances prediction performance and also allows the LLM to provide more detailed explanations based on the structured alignment. Extensive experiments on qualitative and quantitative tasks including 7 datasets demonstrate that **CALMOL** achieves significant performance in predicting molecule interactions involving new molecules. In future, we leave extending our method to protein analysis for further explorations.

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

# A APPENDIX

## A.1 EXPERIMENT SETTINGS

In this section, we introduce our experimental setups in detail with descriptions of used datasets along with baseline models to benchmark the performance of our proposed method.

### A.1.1 DATASETS

In our experiment, 7 diverse datasets are employed, encompassing both drug-drug interaction and solute-solvent interaction tasks. Here we provide a brief overview of the original datasets. Detailed statistical information on the datasets used in this study can be found in Table 1.

**ZhangDDI** (Zhang et al. (2017)). It consists of 548 drugs and 48,548 drug-drug interaction pairs in total, along with multiple types of similarity information between drug pairs.

**ChChMiner** (Marinka Zitnik & Leskovec (2018)). This dataset contains 1,322 drugs and their labeled DDIs, all of which have been extracted from official drug labels and validated through scientific research.

**DeepDDI** (Ryu et al. (2018)). It collects 1704 various drugs with their labeled DDIs. The collection is gathered from DrugBank which features detailed DDI data alongside associated side-effect annotations.

**FreeSolv** (Mobley & Guthrie (2014)). The dataset includes 643 hydration free energy measurements for small molecules in water, both experimental and calculated. For our study, we focus on 560 experimental values, consistent with previous work.

**CompSol** (Moine et al. (2017)). This dataset aims to demonstrate the influence of hydrogen-bonding interactions on solvation energies. It includes a total of 3,548 combinations involving 442 unique solutes and 259 solvents, as referenced in earlier studies.

**Abraham** (Grubbs et al. (2010)). It compiles information published by the Abraham research group at University College London. It includes 6,091 combinations of 1,038 unique solutes and 122 solvents, in accordance with prior studies.

**CombiSolv** (Vermeire & Green (2021)). It integrates data from the MNSol, FreeSolv, CompSol, and Abraham datasets, resulting in 8780 unique pairings between 1,415 solutes and 309 solvents

### A.1.2 BASELINES

In this section, we provide introduction of the baseline models utilized in our experiment. Both traditional deep learning based methods and the recent biochemical LLMs are employed. For qualitative tasks, we use the following baselines:

**CIGIN** (Pathak et al. (2020)). This model uses a three-phase framework—message passing, interaction, and prediction—to achieve high accuracy in solvation free energy predictions and provides chemically interpretable insights into electronic and steric factors governing solubility.

**SSI-DDI** (Nyamabo et al. (2021)). This method applies a 4-layer GAT model to uncover substructures across different layers, while the co-attention mechanism handles the final prediction.

**DSN-DDI** (Li et al. (2023b)). It persents a dual-view drug representation learning network that integrates local and global drug substructure information from both individual drugs ('intra-view') and drug pairs ('inter-view').

**CMRL** (Lee et al. (2023b)). The approach reveals the main substructure driving chemical reactions through a conditional intervention model that adapts its intervention based on the paired molecule.

**CGIB** (Lee et al. (2023a)). It adapts the detected substructure depending on the paired molecule to mimic real chemical reactions, based on the conditional graph information bottleneck theory.

**DeepDDI** (Ryu et al. (2018)). In this method, the structural similarity profile of the two drugs is first evaluated against other drugs, after which a deep neural network is used to complete the prediction.

**MHCADDI** (Deac et al. (2019)). The model utilizes a gated information transfer neural network to manage substructure extraction, and interactions are guided by an attention mechanism.

**MolTC** (Fang et al. (2024)). It introduces a novel multi-modal framework that integrates molecular graph structures and LLMs using Chain-of-Thought (CoT) theory.

As for quantitative tasks, the following baselines are employed besides CIGIN, CMRL, CGIB and MolTC which are mentioned above:

**D-MPNN** (Vermeire & Green (2021)). This technique combines the fundamentals of quantum calculations with the experimental precision of solvation free energy measurements, using a transfer learning approach with the CombiSolv-QM and CombiSolv-Exp databases.

## B  ABLATION STUDY

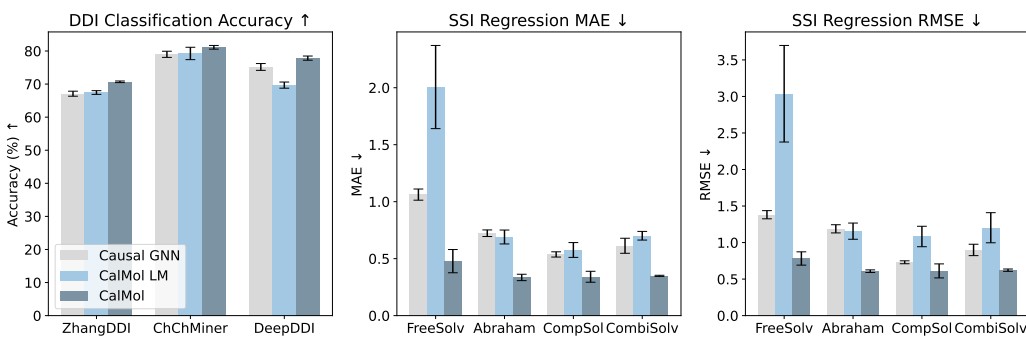

Figure 4: Ablation study.

The ablation studies we conducted are shown in Figure 4. A detailed evaluation was performed to compare our approach with the standalone use of Causal Molecule Motif-interaction Disentangling module proposed in Section 3.1, and the solely LM-based **CALMOL** without Causal GNN, across diverse task settings and datasets. It is evident that full **CALMOL** excels in the context of both DDI classification and SSI regression tasks on a wide range of datasets. The most striking difference can be seen in quantitative SSI tasks, where its superiority are most pronounced.

