# OpenReview forum: "CaLMol: Disentangled Causal Graph LLM for Molecular Relational Learning"
_ICLR.cc/2025/Conference — Submitted to ICLR 2025_

### Official Review · Reviewer_Y1vr · 2024-10-31

**Soundness:** 3
**Presentation:** 2
**Contribution:** 2
**Rating:** 3
**Confidence:** 4

**Summary:**

This paper proposes a method to keep the invariant between molecular structures and semantic texts under a zero-shot scenario. The topic is interesting, and the experimental results look positive. Unfortunately, the paper is vague and lacks clarity both in the description of the technical approach and in the construction of the proposed datasets used for training.

**Strengths:**

The topic is valuable and interesting. Introducing functional substructures based on LLM makes it intuitive to predict potential molecular interactions.

**Weaknesses:**

1. How to introduce supervised signals to optimize the weights between motifs from different molecules is confusing, and it is suggested that the authors provide more details to clarify the principles of calculating the weights between motifs, and what the symbol \hat{Y}_C, \hat{Y}_S, \hat{Y} stand for.

2. The core idea of CalMol is similar to MolTC[1], the authors should clarify the key difference between them.

3. The ablation study is limited, the authors should further discuss the contribution of the LLM backbone. Besides, the contribution of casual GNN is weak in the DDI prediction task, but it shows strong promotion on SSI prediction, the authors can discuss this phenomenon.

[1] Fang, J., Zhang, S., Wu, C., Yang, Z., Liu, Z., Li, S., ... & Wang, X. (2024). Moltc: Towards molecular relational modeling in language models. arXiv preprint arXiv:2402.03781.

**Questions:**

see the weaknesses.

---

### Official Review · Reviewer_9ZZt · 2024-10-31

**Soundness:** 3
**Presentation:** 3
**Contribution:** 3
**Rating:** 3
**Confidence:** 5

**Summary:**

This paper introduces CALMOL, a causally disentangled invariant graph large language model (LLM) tailored for molecular relational learning (MRL), with a particular focus on zero-shot scenarios requiring predictions of new molecular interactions. By integrating Graph Neural Networks (GNNs) with LLMs, CALMOL captures causal structural relationships and aligns molecular structures with semantic information, thereby improving predictions in drug design and molecular interaction studies. Overall, this paper is highly intriguing and meaningful, but there are several issues that require attention.

**Strengths:**

1. The starting point of this paper is interesting; exploring causal substructures with large models is indeed an engaging and meaningful topic.
2. Generalization and Robustness: By leveraging invariant relationships across molecular structures and text, CALMOL effectively addresses distribution shifts between known and new drugs, thus enhancing generalization to unseen molecules. CALMOL maintains consistent performance across various dataset splits (Section 4.1).

**Weaknesses:**

1. **Assumption on Molecular Distributions**: The paper claims that most existing MRL methods assume the same molecular distributions. However, I rarely encounter papers that explicitly make assumptions about molecular distributions, and the term "molecular distributions" is somewhat ambiguous, requiring further clarification. To substantiate this claim, I would recommend that the authors provide specific examples of existing MRL methods that make this assumption or clarify precisely what they mean by "molecular distributions" in this context.

2.  **Effectiveness of Molecular Feature Extraction**: The model only uses SMILES information during the modality alignment process, yet SMILES is also provided in the input. This raises questions about the effectiveness and actual contribution of molecular graph feature extraction. I suggest the authors clarify the role and contribution of molecular graph feature extraction in their model, given that SMILES information is used in multiple stages. An ablation study or analysis showing the added value of graph feature extraction over using SMILES alone would be helpful in addressing this concern.

3. **Novelty of the Method**: The method’s novelty is questionable; the paper seems to merely link motif sets’ causal motif extraction with LLMs in a fairly straightforward manner, without a clear motivation. Additionally, the paper claims that the LLM provides further interpretability, yet no relevant case study is provided in the experimental section to support this. I suggest that the authors provide a more detailed comparison with existing methods that combine causal motif extraction and LLMs, highlighting any specific innovations in their approach. Including a case study or examples demonstrating the enhanced interpretability claimed for their LLM-based approach would strengthen the paper.

4. **Interpretability Challenges**: While CALMOL offers causal substructure explanations, the interpretability of predictions could be improved. Providing more detailed analyses or visual examples would better illustrate how causal substructure disentanglement directly impacts interaction predictions (Section 3.1). This could offer greater clarity on the added interpretability benefits of the model.

5. **Dependency on LLMs**: Due to computational demands, CALMOL’s reliance on large language models may limit its applicability in resource-constrained environments. Furthermore, the paper does not clearly demonstrate any significant advantage of LLMs in this domain. I suggest the authors provide a more detailed discussion of the computational requirements of their model, ideally comparing performance versus computational cost with non-LLM methods. Specific examples or analyses that demonstrate the unique advantages that LLMs bring to molecular relational learning tasks would also help to substantiate this aspect.

**Questions:**

1. Please provide specific examples of existing MRL methods that make this assumption about molecular distributions, or clarify precisely what is meant by "molecular distributions" in this context. Are the authors referring to "element distribution" or "atom distribution"? Providing this clarification will help address the concern more directly and substantiate the authors' claims.


2. The model input includes both the molecular graph information and the SMILES representation; it seems an additional ablation study is needed to demonstrate the effectiveness of both modalities like MolCA .


3. After obtaining the substructure based on causal theory, why is it necessary to input it into a large language model rather than making a direct prediction? Does this approach truly improve the final predictive results? Furthermore, while the manuscript mentions that llm could enhance interpretability, I could not find any experiments or examples to support this claim.


4. With the introduction of a LLM, the model's complexity and resource consumption should be compared with that of conventional models to verify the necessity of incorporating LLMs, allowing for a more comprehensive evaluation.


5. More llm-based model are needed as baseline to verify CALMOL's performance.



[1] MolTC: Towards Molecular Relational Modeling In Language Models；

[2] MolCA: Molecular Graph-Language Modeling with Cross-Modal Projector and Uni-Modal Adapter

---

### Official Review · Reviewer_jTk6 · 2024-11-01

**Soundness:** 2
**Presentation:** 2
**Contribution:** 2
**Rating:** 3
**Confidence:** 3

**Summary:**

This work presents CalMol, a molecular relationship learning framework based on large models and disentanglement. CalMol consists of two main parts: a causal substructure extraction module and a multimodal large model fusion module. The causal substructure extraction module learns the core substructures of molecules by decomposing the target molecule and studying the substructures in contact between pairs of molecules. The multimodal large model fusion module integrates natural language instructions with SMILES and graphical representations of molecules and core substructures into LLM for downstream tasks by constructing prompts. This work is based on MolTC, with the addition of a causal substructure extraction module. The authors evaluated CalMol on DDI (drug-drug interaction) and SSI (solute-solvent interaction) tasks, where CalMol achieved comparative performance on the DDI task and notable performance on the SSI task.

**Strengths:**

This work presents CalMol, a molecular relationship learning framework based on large models and disentanglement, which achieved comparative performance on the DDI task and notable performance on the SSI task. Extracting the causal substructures of molecules is an interesting topic.

**Weaknesses:**

1. The authors believe that existing methods rely on "variant molecular structures", which hinders their performance, but there is a lack of a clear definition of "variant molecular structures".
2. For a molecule, the substructures that play a key role may vary when it binds with different molecules, i.e., the so-called core substructures are not fixed. Therefore, it is not rigorous enough to determine the core substructures of a molecule with just one set of relationships.
3. Using a substructure of a molecule as its causal substructure is somewhat far-fetched, especially for larger molecules.
4. The supervision signal and loss function used in the substructure learning stage are unclear.
5. The authors propose to make the disentangled spurious part S approach a random distribution, but the rationale for doing so is not explained.
6. There is a lack of necessary ablation experiments, such as whether the disentanglement module is effective and whether the several disentanglement losses are necessary.

**Questions:**

As stated in the Weaknesses.

---

### Official Review · Reviewer_Aib3 · 2024-11-03

**Soundness:** 3
**Presentation:** 3
**Contribution:** 2
**Rating:** 5
**Confidence:** 4

**Summary:**

This paper presents CaLMol, a model for molecular relational learning (MRL) that uses a combination of Graph Neural Networks (GNNs) and Large Language Models (LLMs) to predict drug-drug (DDI) and solute-solvent (SSI) interactions in a zero-shot setting. The model’s innovative approach in leveraging causal disentanglement and aligning molecular structures with semantic information provides a promising direction.

**Strengths:**

- This paper combines causal disentanglement and semantic alignment between GNN and LLM, allowing for a comprehensive understanding of molecular interactions.
- By targeting unseen molecules, CaLMol addresses an important area in MRL, providing potential for applications involving new drugs or compounds.
- The model is evaluated across multiple datasets, showing improvements in accuracy over several baselines, which demonstrates its effectiveness in specific zero-shot tasks.
- The paper is well-written and easy to follow.

**Weaknesses:**

See Questions.

**Questions:**

- Could the authors provide additional analysis on the computational complexity of CaLMol? How about the comparison with these baselines in training time and inference time?
- More detail about interpretability cases and analysis should be provided to support the advantage of CaLMol.
- In Table 1, it is evident that the three datasets for DDI classification present a highly imbalanced binary classification task; however, the results shown for CaLMol in Table 2 perform poorly on AUC-ROC, which is a crucial metric for imbalanced data.
- Given the model’s dependency on selected datasets, how would the authors suggest extending the approach to larger and more diverse datasets? For example, Drug-Target Interaction (DTI) is also a significant task in drug discovery; demonstrating that CaLMol is useful in this task would enhance its practical significance.

---

### Author Response · Authors · 2024-12-04
**Summary**

Dear reviewers,

We sincerely thank all the reviewers for dedicating your valuable time and effort to evaluate our work.

We would like to summarize the revised paper **`pdf`** as below:

1. We updated a more concrete `Figure 1`, to illustrate the motivation of our work: MRL is driven by causal substructure pair and related property. The interaction between these two drugs is primarily driven by the imidazole ring in fluconazole, which inhibits the CYP2C9 enzyme responsible for metabolizing the coumarin core in warfarin. This inhibition slows down the breakdown of warfarin, causing its concentration to increase in the bloodstream, which heightens the risk of excessive anticoagulation and bleeding.
2. We have included a more comprehensive ablation study in `Appendix B` to evaluate each component of our model.

Although the rebuttal period is limited, we sincerely hope our responses have addressed your concerns and provided greater clarity about our work. We are committed to further refining our research based on your valuable feedback!

---

### Meta-Review · Area_Chair_yEGL · 2024-12-17

**Metareview:**

### Summary
The paper introduces CaLMol, a model for molecular relational learning (MRL) in a zero-shot setting. It integrates causal disentanglement and semantic alignment between Graph Neural Networks (GNNs) and Large Language Models (LLMs) to predict drug-drug interactions (DDI) and solute-solvent interactions (SSI). The method extracts causal substructures of molecules to enhance generalization to unseen data. Experiments on multiple datasets demonstrate performance improvements, though the contributions remain incremental.

### Strengths
- The paper addresses molecular relational learning in a zero-shot setting, which is both practical and underexplored, particularly for unseen drugs and molecules.
- The idea of extracting functional causal substructures and aligning them with semantic information via LLMs is innovative and adds interpretability.
- Experiments on DDI and SSI tasks demonstrate effectiveness, with CaLMol showing consistent performance improvements across several benchmarks.

### Weaknesses
- Key concepts such as "causal substructures" and "molecular distributions" are poorly defined, and important implementation details (e.g., optimization signals, weight calculations) are missing or unclear.

- The method appears to be an incremental extension of MolTC, combining causal motif extraction and LLMs without substantial innovation. Clear differentiation from related work is lacking.

- Incomplete Ablation and Analysis: There is no clear ablation study to show the individual contributions of the GNN and LLM components.
The model's reliance on LLMs raises concerns about computational efficiency, which is not adequately discussed or compared to simpler methods.

While CaLMol addresses an important and interesting problem in molecular relational learning, the paper suffers from poor methodological clarity, limited novelty, and incomplete experimental analysis. The lack of rigorous ablation studies and unclear differentiation from prior work (e.g., MolTC) weakens the strength of its contributions. Improvements in clarity, theoretical justification, and additional analyses are needed to validate the method's significance.

**Additional Comments On Reviewer Discussion:**

The major concerns raised by the reviewers are:

- Lack of Methodological Clarity: Key concepts, such as causal substructures, molecular distributions, and optimization principles, are not well-defined. Important implementation details (e.g., weight calculations, supervised signals, and disentanglement losses) are unclear or missing, leading to confusion.

- Limited Novelty: The method appears to be an incremental extension of MolTC, combining existing ideas (e.g., causal motif extraction and LLMs) without sufficient innovation. The paper does not clearly differentiate CaLMol from prior work, reducing its perceived contribution.

- Incomplete Experiments and Analysis: The ablation study is insufficient, failing to clarify the contribution of individual components (e.g., LLM vs. GNN). Computational complexity, efficiency, and comparisons to non-LLM baselines are not adequately addressed, raising concerns about practical applicability.

Rather than addressing the reviews point-by-point, the authors uploaded a revised manuscript, which makes it hard to follow which points were indeed addressed. In general, the novelty concerns have not been addressed.

---

### Decision · Program_Chairs · 2025-01-22

Reject